# Frequency-Dependent Neural Modulation of Dorsal Horn Neurons by Kilohertz Spinal Cord Stimulation in Rats

**DOI:** 10.3390/biomedicines12061346

**Published:** 2024-06-18

**Authors:** Dong Wang, Kwan Yeop Lee, Zachary B. Kagan, Kerry Bradley, Dongchul Lee

**Affiliations:** Nevro Corporation, Redwood City, CA 94065, USA

**Keywords:** spinal cord stimulation, pain, 10 kHz frequency, calcium imaging, superficial dorsal horn

## Abstract

Kilohertz high-frequency spinal cord stimulation (kHF-SCS) is a rapidly advancing neuromodulatory technique in the clinical management of chronic pain. However, the precise cellular mechanisms underlying kHF-SCS-induced paresthesia-free pain relief, as well as the neural responses within spinal pain circuits, remain largely unexplored. In this study, using a novel preparation, we investigated the impact of varying kilohertz frequency SCS on dorsal horn neuron activation. Employing calcium imaging on isolated spinal cord slices, we found that extracellular electric fields at kilohertz frequencies (1, 3, 5, 8, and 10 kHz) induce distinct patterns of activation in dorsal horn neurons. Notably, as the frequency of extracellular electric fields increased, there was a clear and significant monotonic escalation in neuronal activity. This phenomenon was observed not only in superficial dorsal horn neurons, but also in those located deeper within the dorsal horn. Our study demonstrates the unique patterns of dorsal horn neuron activation in response to varying kilohertz frequencies of extracellular electric fields, and we contribute to a deeper understanding of how kHF-SCS induces paresthesia-free pain relief. Furthermore, our study highlights the potential for kHF-SCS to modulate sensory information processing within spinal pain circuits. These insights pave the way for future research aimed at optimizing kHF-SCS parameters and refining its therapeutic applications in the clinical management of chronic pain.

## 1. Introduction

Spinal cord stimulation (SCS) has been an effective clinical tool for pain management for over 50 years [1]. High frequency (HF) SCS has been widely adopted in the last decade, predominantly because paresthesia-free 10 kHz SCS was shown to achieve statistically and clinically superior pain relief compared to traditional paresthesia-based 40–50 Hz low frequency (LF) SCS [2,3]. Kilohertz HF-SCS involves the application of electrical impulses to the spinal cord, which do not activate the axons of the dorsal columns while effectively modulating the neural circuits involved in pain perception in the dorsal horn [4,5,6]. Ten kHz SCS has been the most-studied frequency, and has demonstrated several distinct characteristics: unlike LF-SCS, 10 kHz SCS, as applied in clinical practice, does not produce paresthesia, making it a more comfortable and patient-friendly option [3,7]. Moreover, 10 kHz SCS has shown promise in achieving sustained pain relief without the risk of habituation or tolerance, which can occur with opioid-based pain management strategies [8,9]. The results of clinical trials also demonstrate the effectiveness of 10 kHz SCS in treating a wide spectrum of chronic pain conditions, including diabetic neuropathic pain [10], failed back surgery syndrome (FBSS) [3], and complex regional pain syndrome (CRPS) [11].

The mechanism of action underlying kHF-SCS is complex and not yet fully elucidated, but is believed to involve the recruitment of inhibitory pathways within the spinal cord [5,12,13]. Recent studies using a rat model of neuropathic pain suggested that kHF-SCS may promote the release of endogenous opioids as part of its pain-relieving mechanism [14] or potentially reduces hyperalgesia by enhancing autophagic activity within the spinal cord [15]. Moreover, kHF-SCS may modulate spinal cord neuronal excitability and synaptic transmission, ultimately leading to a reduction in pain signals reaching the brain [16].

Stimulation of spinal cord dorsal horn neurons through kHF-SCS is a key mechanism in pain management. The dorsal horn of the spinal cord is integral to processing pain signals, where sensory information from peripheral nerves is transmitted to higher brain centers [17]. By targeting specific regions within the dorsal horn, kHF-SCS can inhibit the transmission of nociceptive signals while promoting the activation of inhibitory pathways, ultimately leading to pain relief. Understanding the precise effects of kHF-SCS on dorsal horn neuron activation is crucial for optimizing treatment outcomes and developing more effective pain management strategies for individuals with chronic pain conditions. This study seeks to offer a comprehensive comparison of neuron activation induced using kilohertz-range SCS, specifically exploring frequencies ranging from 1 kHz to 10 kHz. Our findings reveal that, in contrast to lower kilohertz SCS frequencies, 10 kHz SCS elicits greater activation of dorsal horn neurons, particularly within the deeper laminae of the dorsal horn regions.

## 2. Materials and Methods

### 2.1. Animals

All experimental procedures were conducted in accordance with animal care guidelines approved by the Institutional Animal Care and Use Committee (IACUC) at Explora BioLabs (San Diego, CA, USA) (#EB20-016-001, 11 January 2021). Healthy *Sprague Dawley* rat pups, aged between 7 and 12 days (body weight 12–18 g), were used in our experiments. These pups were housed with their mother, and were maintained on a 12 h light/dark cycle in a temperature-controlled environment with unrestricted access to food and water. Typically, male rat pups were selected.

### 2.2. Spinal Cord Slice Preparation

The procedure for spinal cord slice preparation closely followed the methods previously described [18]. Briefly, rat pups aged between 7 and 12 days were anesthetized using isoflurane, decapitated, and their vertebral columns were rapidly removed and placed in oxygenated ice-cold dissection solution (in mM: 95 NaCl, 2.5 KCl, 1.25 NaH_2_PO_4_, 26 NaHCO_3_, 50 sucrose, 25 glucose, 6 MgCl_2_, 1.5 CaCl_2_, and 1 kynurenic acid, pH 7.4, 320 mOsm). The isolated lumbar spinal cord was embedded in a 3% agarose block, and transverse slices of 400 µm thickness were generated using a vibrating microtome (Leica VT1000S). Subsequently, the slices were incubated in an oxygenated recovery solution (in mM: 125 NaCl, 2.5 KCl, 1.25 NaH_2_PO4, 26 NaHCO_3_, 25 glucose, 6 MgCl_2_, and 1.5 CaCl_2_, pH 7.4, 320 mOsm) containing a fluorogenic calcium-sensitive dye (5 µM, Aat Bioquest Calbryte™ 520 AM, Cat# 20653) for 30 min at 35 °C [19]. Slices loaded with Calbryte™ 520 AM were transferred to the recovery solution at room temperature prior to the imaging study.

### 2.3. Calcium Imaging with Kilohertz SCS

Calcium imaging was conducted using an Olympus BX51WI microscope (Waltham, MA, USA) equipped with a high-speed Hamamatsu ORCA-Flash4.0 V3 Digital CMOS camera (model# C13440-20CU). Slices were transferred to the recording chamber and perfused at approximately 2 mL/min with oxygenated recording solution (recovery solution containing 1 mM MgCl_2_ and 2 mM CaCl_2_).

A Nevro trial stimulator (TSM) was used to generate extracellular electric fields at varying frequencies (1 kHz, 3 kHz, 5 kHz, 8 kHz, 10 kHz, and 1 kHz repeated) with a consistent pulse width of 30 µs (charge-balanced biphasic square pulses). The stimulation pulse amplitude was adjusted to 6 mA in order to achieve detectable fluorescence change in the full range of frequencies we tried. The stimulation was applied to the slices in ascending (or descending) frequency order using a platinum microelectrode (WPI, Sarasota, FL) in artificial cerebrospinal fluid (aCSF) with a platinum wire (>3 mm immerged in the aCSF solution) as a return electrode. The stimulation electrode was positioned approximately 200 µm from the outer circumference of the slice to simulate an epidural placement (Figure 1A). The return electrode was placed along the curvature of the ventral area of the slice. After recording a 10 s no-stimulation baseline, each frequency was delivered for 10 s. Images were acquired using the Olympus CellSen software (V4.2) with a sampling rate of 2 Hz (Figure 1B) and a resolution of 2048 × 2048 under a 40× magnification.

### 2.4. Imaging Data Analysis and Statistics

Image data were processed and analyzed using custom MATLAB (Mathworks, Naticks, MA, USA) functions combined with an image processing toolbox. Individual cells were marked manually and pixels (5 µm × 5 µm) around the marked point were used for fluorescence data. Fluorescence changes (dF) resulting from stimulation (Figure 1C) were computed by determining the difference between the baseline F_0_ (average of the first 10 s) and individual frames (normalized by F_0_) during stimulation (Figure 1D). To mitigate the impact of stimulation sequence, the frequency sequence was applied in a descending order (1, 10, 8, 5, 3, and 1 kHz) for half of the samples and ascending order (1, 3, 5, 8, 10, and 1 kHz), with the exception of 1 kHz stimulation, which was consistently applied at the beginning and end of the sequence. This served as a reference to evaluate the repeatability of neural responses. The initial step involved employing the Nonparametric Kruskal–Wallis one-way analysis of variance (ANOVA) to assess significance (*p* < 0.01) among groups. Subsequently, a post hoc two-sided Wilcoxon rank sum test, adjusted for the Bonferroni correction (*p* < 0.01), was utilized to determine the significance of differences between groups.

## 3. Results

### 3.1. Kilohertz SCS Elicits Activation of Dorsal Horn Neurons

Across all the tested stimulation frequencies, we observed marked changes in fluorescence intensity, mediated by the calcium-sensitive dye Cal-520.

To ensure the reliability and reproducibility of dorsal horn neuron responses, we implemented two criteria to include a sequence of trials in analysis: (1) a minimum dF (change in fluorescence) of greater than 0.2% for both the initial and repeated 1 kHz trials, and (2) a second trial dF within 20% to 120% of initial 1 kHz trial dF. The defined variance was empirically selected based on the general response of neurons to 1 kHz over time. Among the total neurons assessed (*n* = 520, gray dots in Figure 2A), approximately half of these neurons (*n* = 230, black dots) met these criteria. This suggests that a specific subpopulation of dorsal horn neurons consistently and reliably responds to SCS, even when subjected to different stimulation frequencies.

Upon closer examination of the chosen neurons meeting the specified criteria and subjected to 1 kHz spinal cord stimulation (SCS), it was observed that more than 90% of them exhibited fluorescence changes falling within the range of 5% to 20% (Figure 2B). Notably, the first trial exhibited a slightly stronger response of dF compared to the final trial. This discrepancy may be attributed to a reduction in sensitivity resulting from multiple stimulations at varying frequencies or the gradual photo-bleaching effect over time that is known to occur when using this dye [20]. For the purpose of frequency comparisons in subsequent analyses, we utilized the first trial of 1 kHz SCS as a reference point against other frequencies.

### 3.2. Frequency-Dependent Activation of Dorsal Horn Neurons by SCS

Our results demonstrate a frequency-dependent response pattern in dorsal horn neuron activation. The distribution of neuron responses exhibited a clear ascending order relative to the stimulation frequency (Figure 3A). Notably, while most neurons activated by 1 kHz SCS demonstrated changes in fluorescence intensity (dF) of less than 20%, more than 50% of neurons subjected to 10 kHz SCS displayed a substantial dF change exceeding 20%. The distribution of dF from 8 kHz SCS closely resembled that of 10 kHz SCS responses. Neurons with initially weak responses (<10%) during 1 kHz stimulation exhibited an increase in their responses with 5 kHz, further increasing with 8 kHz and 10 kHz stimulation frequencies.

Quantitative analysis (Figure 3B) revealed that all parameters (mean ± standard deviation and median values) consistently increased with higher stimulation frequencies. Specifically, the mean ± SD and median values of dF reached their maxima at 10 kHz, with values of +21% ± 14%, 16%, when compared to 8 kHz (+20% ± 12%, 16%), 5 kHz (+16% ± 10%, 14%), 3 kHz (+13% ± 9%, 10%), and 1 kHz (+9% ± 7%, 7%). It is noteworthy that the dF for the repeated 1 kHz stimulation at the end of the sequence was approximately 32% less than the dF for the initial 1 kHz stimulation.

The sequence using ascending stimulation frequencies was used for three rats, whereas descending frequencies were used for two. Individual neuron data (depicted as dots) for each frequency (columns) with mean animal dF trended upwards with increasing frequency (Figure 3C). Order effects, such as photo-bleaching or sensitivity reduction over time and after multiple stimulations, were likely contributors to these variations. However, even with these order effects, the average data from all five rats consistently demonstrated that kilohertz SCS modulates dorsal horn neuron activities in a frequency-dependent manner (*p* < 0.01 between 10 kHz and other frequencies, except 8 kHz).

These findings suggest that dorsal horn neuron responses to extracellular electrical fields are frequency dependent, with a stronger effect seen at higher kHz frequencies.

### 3.3. Activation of Deeper Dorsal Horn Neurons by Higher Frequency SCS

We observed that varying frequencies of spinal cord stimulation activated different populations of dorsal horn neurons, shedding light on depth-specific responses. Notably, the selected neurons by reproducibility criteria, spanned a range of depths, extending from 50 to 350 µm beneath the dorsal horn’s surface. On average, neurons located between 100 to 300 µm in depth were distributed uniformly across the five rats, with minor variations (Figure 4A).

To better understand the depth-specific effects, we plotted each neuron’s response to 1 kHz, 5 kHz, and 10 kHz SCS individually (Figure 4B–D). Across all depths, a clear trend emerged: higher frequencies consistently activated a greater number of dorsal horn neurons.

Depth analysis demonstrated that neurons located closer to the surface (<100 µm) were more effectively activated compared to their deeper counterparts (up to 350 µm) (Figure 4E) for all frequencies. This depth-dependent response can be attributed to the phenomenon of electric field decay, as it penetrates deeper into neural tissue.

Furthermore, the median fluorescence changes for 10 kHz SCS, in comparison to 1 kHz SCS, were notably higher at shallower depths, with changes of 27% for 10 kHz versus 9% for 1 kHz at depths less than 150 µm and 20% for 10 kHz versus 8% for 1 kHz at 200 µm. Importantly, the difference in response between 10 kHz and 1 kHz SCS was statistically significant (*p* < 0.01) up to a depth of 300 µm. These findings indicate that higher kHz SCS activates more and deeper dorsal horn neurons than lower kHz frequencies.

### 3.4. Neuronal Fluorescence Increases More in Response to 10 kHz SCS Compared to Lower kHz Frequencies

Next, we analyzed the individual neuron responses to various frequencies compared to 10 kHz SCS (Figure 5). Neurons responded very similarly to both 8 kHz and 10 kHz, with only a few neurons exhibiting slightly stronger responses to 10 kHz (Figure 5A). This was less true for lower kHz frequencies. A ‘strong 10 kHz responder’ analysis, defined as neurons displaying a response greater than 40% in 10 kHz (as indicated by circles in Figure 5C,D), were less activated in response to 1 kHz and 3 kHz stimulation.

These findings underscore the stronger neuronal responses to 10 kHz SCS in comparison to other kilohertz frequencies.

### 3.5. Preference for Higher Frequencies in Activating Dorsal Horn Neurons across Depths

To gain a deeper understanding of the frequency preference exhibited by individual neurons, we conducted further analysis. In comparing the responses to 1 kHz and 10 kHz SCS, we studied frequency preference based on the relative location of data points in the plot (Figure 6A). Neuronal responses were categorized into three areas of frequency preference: (1) those that activated most by 1 kHz, (2) those activated similarly by 1 kHz or 10 kHz, and (3) those that activated most by 10 kHz.

This frequency preference was then analyzed in relation to the depth of neurons (Figure 6B,C). For the majority of neurons spanning depths from 50 to 300 µm, over 90% exhibited a mild or high preference for 10 kHz stimulation. Notably, superficial neurons, particularly those close to the surface, displayed a pronounced preference for 10 kHz, with this preference decreasing as depth increased, transitioning to a mild preference for 10 kHz. Conversely, only a small fraction (<10%) of neurons displayed a preference for 1 kHz at any depth.

These findings underscore a consistent and robust preference for higher frequencies, particularly 10 kHz, in activating dorsal horn neurons across various depths. The transition from high to mild preference for 10 kHz with increasing depth suggests that superficial neurons are especially receptive to higher frequencies.

### 3.6. Strongest Dorsal Horn Neuron Activation by 10 kHz and 8 kHz SCS

To identify the optimal frequency for eliciting the greatest response in all selected neurons, we conducted an analysis of ‘preferred frequency: the frequency made highest responses’ by depth (Figure 7). Between 0 and 250 µm deep, 90% of neurons exhibited their greatest responses when subjected to either 10 kHz or 8 kHz SCS (Figure 7A). At a depth of 100 µm (*n* = 66 neurons), the preference for 10 kHz was prominent, with 34 (52%) neurons exhibiting the highest response, whereas 16 (24%) neurons had their largest response to 8 kHz. Conversely, only one neuron showed a preferential response to 1 kHz. This trend persisted deeper into the spinal cord, with 88% of neurons at 200 µm depth (*n* = 45) demonstrating a preference for either 10 kHz or 8 kHz. Only two neurons at this depth were activated most effectively by 1 kHz.

Importantly, this frequency preference was consistent across all depths, with the percentage of neurons exhibiting 1 kHz as the best frequency remaining low (4%), except for a few neurons at specific depths.

Furthermore, the majority of neurons (>70%) demonstrated their highest activation with either 8 kHz or 10 kHz at all depths, up to 250 µm. Neurons located beyond a depth of 250 µm were excluded from consideration due to insufficient activation. The relative populations that responded optimally to 8 kHz or 10 kHz reached their maximum at depths of 150 µm and 200 µm, comprising 85% and 89% of the neuron population, respectively (Figure 7B).

These findings underscore the robust and consistent activation of dorsal horn neurons by 10 kHz and 8 kHz SCS at all depths of the dorsal horn.

## 4. Discussion

The outcomes of our investigation offer valuable insights into the impact of kilohertz SCS on dorsal horn neurons, unveiling patterns of frequency-dependent activation and depth-specific responses. To our knowledge, this is the first study of the effects of kilohertz SCS in isolated rat spinal cord slices, and we believe this work contributes novel perspectives to the existing body of knowledge.

### 4.1. Frequency-Dependent Activation of Dorsal Horn Neurons

Our study demonstrates a significant influence of kilohertz SCS on the activation of dorsal horn neurons. Previous studies have suggested that these neurons responded to a range of stimulation frequencies [6]. Here, we observed a consistent trend towards greater activation as the kilohertz frequency increased.

Neurons activated using 1 kHz SCS typically exhibited modest changes in dF, with the majority (93%) showing changes below 20%. In contrast, over half of the neurons subjected to 10 kHz SCS displayed dF changes exceeding 20%. This increase in neural response suggests that higher kilohertz frequencies are a more powerful tool for the modulation of dorsal horn neurons in SCS.

Further exploration of the specific effects (e.g., neural, end-organ) of different frequencies is warranted to determine the optimal parameters for various applications to maximize the therapeutic potential of kilohertz SCS.

### 4.2. Depth-Specific Activation of Dorsal Horn Neurons

We observed that higher-frequency SCS, particularly at 8 kHz and 10 kHz, were more effective in activating neurons situated deeper within the dorsal horn. Certainly, depth-specific activation is partially attributed to the reduction in the electric field strength, as it penetrates deeper into the neural tissue, but such effects would be relevant for all frequencies. The mechanism behind this phenomenon may link to the ability of higher frequency electrical fields to effectively modulate neural circuits across a broader spatial range. It is possible that 8 kHz and 10 kHz alter the way electrical signals propagate through neural structures and modulate synaptic transmission in a manner that selectively influences neurons in deeper dorsal horn laminae. This enhanced activity at high kilohertz (e.g., >5 kHz) may not be a simple post-synaptic temporal summation by generating action potentials from axons because the highest firing rate of action potential cannot follow (or saturate) 5, or even 1 kHz. There must be mechanisms involved to integrate small changes made by individual biphasic pulses, not discrete information like action potentials. Our findings provide evidence that high-kilohertz stimulation on central nervous systems may generate action potentials, rather than block action potential propagation, as observed in peripheral nervous systems.

The ability of higher-kilohertz SCS to achieve deeper dorsal horn neuron activation may have important clinical implications. Deeper neurons are often associated with distinct pain signaling pathways [21,22,23], and the ability to drive these neurons using 10 kHz SCS may provide a rational basis for the observed clinically and statistically superior relief of pain using 10 kHz SCS compared to lower frequency SCS [2,3]. We observed that 10 kHz significantly activated more neurons than 1 kHz SCS up to a depth of 300 μm, which underscores the potential of 10 kHz SCS for modulating deeper neural circuits. Further research is required to elucidate the precise mechanisms and therapeutic applications of this deeper activation.

Some neurons exhibited two to three times greater changes in fluorescence compared to 1 kHz stimulation (Figure 6 and Figure 7). This observation aligns with the work by Lee et al., which demonstrated that 10 kHz stimulation induced strong responses in inhibitory neurons, while excitatory neurons exhibited only spontaneous firing [5]. Notably, our findings here indicate that these strong responders, potentially inhibitory neurons, were not selectively activated more than other neurons in response to 1 kHz stimulation. This underscores the potential of 10 kHz SCS to selectively target inhibitory neurons, suggesting its capacity to provide more effective pain relief.

### 4.3. Limitations of the Study

One key limitation of our study is the inability to identify specific spinal cord neuron types that respond to kilohertz SCS. Using transgenic animals expressing genetically encoded calcium indicators [24,25], in particular neuron types such as excitatory or inhibitory neurons, could have provided a more nuanced understanding of how different neuron populations are affected by kilohertz SCS. This is essential for pinpointing the exact neural circuits involved in pain modulation and other functions, as the spinal cord dorsal horn contains a heterogeneous population of neurons with distinct functions, including sensory processing, motor control, and modulatory functions [21,23]. Without neuron type identification, we cannot ascertain whether kilohertz SCS predominantly activates neurons involved in pain perception, motor control, or other functions. This limits our ability to draw specific conclusions regarding the mechanisms of pain relief and neural modulation. In addition, rat pups display a limited rudimentary pain response system and have a less developed spinal cord pain circuitry. While they do exhibit nociceptive behaviors, their neural networks are still maturing, and the connections between neurons are not as intricate as in adults. This immaturity leads to a less sophisticated and potentially less intense experience of pain. Future studies might utilize adult transgenic animal models to explore the specific neuron types activated by kilohertz SCS and their roles in pain modulation. To further the understanding of whether the stimulation effect was direct (via cell depolarization) or indirect (via presynaptic terminal or afferent axon activation), detailed pharmacological studies should be conducted. Our preliminary unpublished data showed that blocking AMPAR and NMDAR using CNQX + AP5 causes a partial reduction in the stimulation effect of 10 kHz SCS. Further studies should be conducted to quantify the contribution of direct vs. indirect effects.

Another limitation was our focus on varying the single parameter of frequency at single pulse width in driving dorsal horn neuron activation. Other literature has attributed differential effects of kilohertz frequency to an increased rate of stimulation charge delivered [26]. Also, as the stimulation frequency is reduced, the pulse width may be increased, and some authors have suggested an equivalency of neural and clinical effects when charge is equilibrated, no matter how it was delivered (e.g., lower frequency and higher pulse width or higher frequency and lower pulse width) [27]. While various studies [28,29] have examined the impact of different pulse widths on dorsal column axons, research specific to dorsal horn neurons is limited. Future work should add in the factor of varied pulse width and frequency to observe whether the same results might be obtained with lower kilohertz frequencies with higher pulse widths.

Along these lines, another means of increasing stimulation charge delivery is to increase the stimulation amplitude. Thus, it is reasonable to question whether adjusting the amplitude of 1 kHz stimulation could achieve an effect equivalent to that of 10 kHz. In this study, we employed a judicious stimulation amplitude of 6 mA to attain the initial detectable fluorescence changes at 1 kHz, and consistently applied this amplitude across all tested frequencies. This amplitude is incomparable to clinically used amplitude because of limited conditions such as spinal cord shape (whole vs. slice), conductive mediums, and electrode geometry. Current flow direction and density will be also different between clinically implanted lead vs. this rodent slice preparation on dish. However, frequency effect may work equally for neurons on top of augmented effect of intact circuits clinically. And the pronounced contrast in neuron response between 1 kHz and 10 kHz underscores the significance of frequency as a crucial element for modulating dorsal neurons under clinically relevant stimulation amplitudes.

Compared to our previous work [5] showing almost no activation at 1–5 kHz, there was a noticeable response in the current study. The main difference might be the previous paper used in vivo preparation where all rostro-caudal connections were intact, while the in vitro slice preparation from this work sacrifices those connections from crucial inhibitory neurons (ex. islet cells with long rostro-caudal innervation) could modulate neighbor neurons [21].

This study could not confirm if 10 kHz activates the dorsal column or not on the transverse slice, while Sagalajev et al. [30] suggested that kilohertz frequency can activate axons at 50% of motor threshold, which is considered to be above the perception threshold in rats [31,32]. Data from our clinical studies have suggested that therapeutic amplitudes for kHz frequency SCS are no greater than 25% of the kHz paresthesia threshold. Thus, the conclusions of Sagalajev et al. relate to paresthesia-based kHz SCS, which is not used clinically.

### 4.4. Clinical Relevance

These findings from the study may have significant implications for the translation of kilohertz SCS from animal models to human clinical applications, particularly in the management of chronic pain and the treatment of various neurological disorders [2,10]. Kilohertz SCS, when applied with an understanding of its frequency-dependent effects, can potentially offer a more precise and patient-specific approach to pain management, e.g., certain patients may respond better to 8 kHz rather than 10 kHz. This tailored approach holds promise for improving patient outcomes, reducing adverse effects, and enhancing the overall efficacy of SCS as a therapeutic modality.

In animal models, the observation that higher kilohertz frequencies can activate deeper dorsal horn neurons suggests that kilohertz SCS may be able to target the distinct pain signaling pathways associated with deeper neural circuits [33,34]. This depth-specificity provides a unique opportunity to address complex and refractory pain conditions that may not be effectively managed with conventional SCS techniques. By modulating deeper neurons, kilohertz SCS may offer relief to patients who have previously found limited success with other treatments.

Translating these findings to human clinical settings open up new avenues for tailoring SCS therapy to individual patient needs. For example, the understanding that certain patients may respond better to specific kilohertz frequencies, such as 8 kHz rather than 10 kHz, highlights the importance of personalized treatment approaches. This tailored approach holds promise for improving patient outcomes, reducing adverse effects, and enhancing the overall efficacy of SCS as a therapeutic modality.

By bridging the gap between animal research and clinical practice, these insights into the frequency-dependent effects of kilohertz SCS offer hope for more effective pain management and treatment of neurological disorders in human patients. Further research and clinical trials will be essential to validate these findings and optimize the implementation of kilohertz SCS in diverse patient populations.

## 5. Conclusions

This study sheds light on the frequency-dependent nature, as well as total numbers and depths of activation, of dorsal horn neurons. Within the kilohertz frequency (1 to 10 kHz) range of field stimulation, a notable trend emerged: higher frequencies correlated with increased activation of dorsal horn neurons, in contrast to blocking. It is noteworthy that the heightened activation associated with higher-frequency spinal cord stimulation persisted even in neurons located at deeper depths (up to 250 µm). Any modulation of neural responses within the spinal network influenced by frequency may have implications for the processing of sensory or pain-related information.

Building upon the findings of this study, further investigation is warranted to elucidate the specific mechanisms underlying the observed correlation between higher stimulation frequencies (1 to 10 kHz) and increased activation of dorsal horn neurons. Additionally, exploring how different frequencies influence the processing of sensory and pain-related information within the spinal network could provide valuable insights into optimizing kHZ-SCS parameters for enhanced pain relief. Moreover, investigating the long-term effects of frequency-dependent modulation on dorsal horn neuron activity and the potential for neuroplastic changes in response to chronic kHZ-SCS therapy would be instrumental in refining treatment strategies for chronic pain management. Overall, future research directions should aim to translate these findings into clinically relevant interventions that improve the efficacy and durability of kHZ-SCS-based therapies for individuals suffering from chronic pain conditions.

## Figures and Tables

**Figure 1 biomedicines-12-01346-f001:**
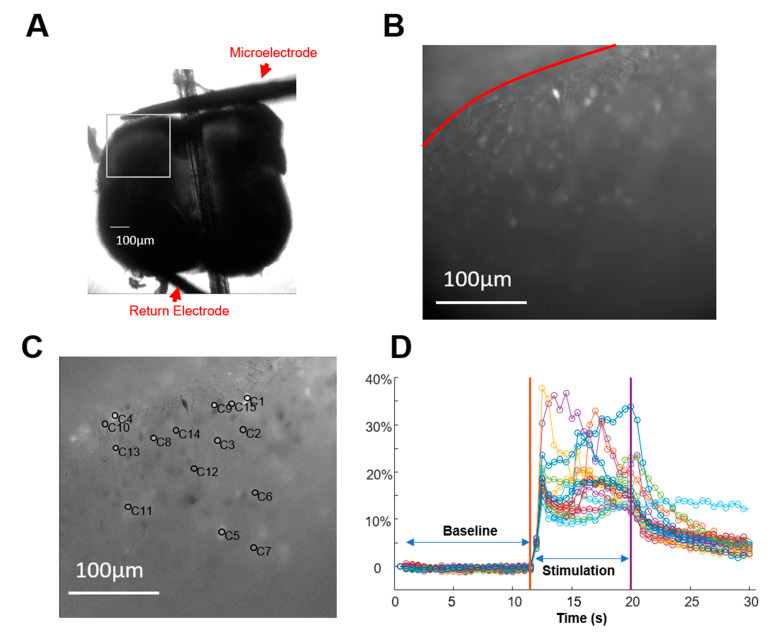
Experimental setup and fluorescence image analysis. (**A**) Spatial configuration of the spinal cord slice with a microelectrode positioned for the delivery of electrical pulses. (**B**) Fluorescence image focused on the square area in (**A**). Fluorescence was recorded for 10 s as a baseline measurement without stimulation. Red solid line indicates the border of the lamina I of the dorsal horn. (**C**) Calculation of fluorescence difference (dF) between the baseline (**B**) and an image captured during stimulation. Neurons of interest were demarcated by circles. (**D**) Plot of dF for each neuron (with different color) after detrending fluorescence over time to account for photo-bleaching. The initial 10 s served as the baseline for computing the effect of stimulation during the subsequent ~10 s (approximately from 10 to 20 s). The response was quantified as the average dF during the stimulation window.

**Figure 2 biomedicines-12-01346-f002:**
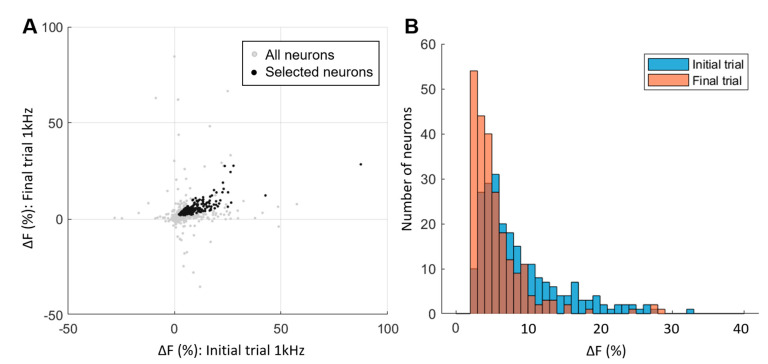
Repeatability, neuron selection, and dF distribution. (**A**) Plot of fluorescence change (dF) for all neurons (*n* = 520) from the initial and repeated trials of 1 kHz stimulation. Neurons exhibiting consistently reproducible dF responses, showcasing a linear correlation between the initial and final trials, were selected for subsequent analysis (indicated in black). (**B**) Distribution of dF values for the selected neurons (*n* = 230) during the initial and final repeated trials. It is noteworthy that over 90% of these neurons exhibited dF changes of less than 20%, and the final trial displayed a decrease in dF for the selected neurons.

**Figure 3 biomedicines-12-01346-f003:**
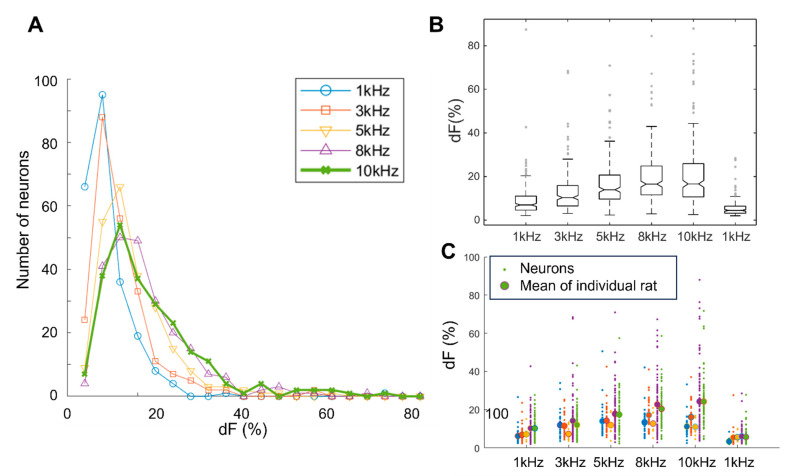
Distribution of dF in response to different frequencies. (**A**) Fluorescence change (dF) in response to various frequencies ranging from 1 kHz to 10 kHz. (**B**) Comparative analysis of dF responses to different frequencies. The repeated response to 1 kHz was plotted after 10 kHz stimulation, demonstrating an overall increase in dF with higher frequencies. (**C**) Individual neuron data (indicated by dots) from each individual rat (presented in separate columns), showing the range of dF responses.

**Figure 4 biomedicines-12-01346-f004:**
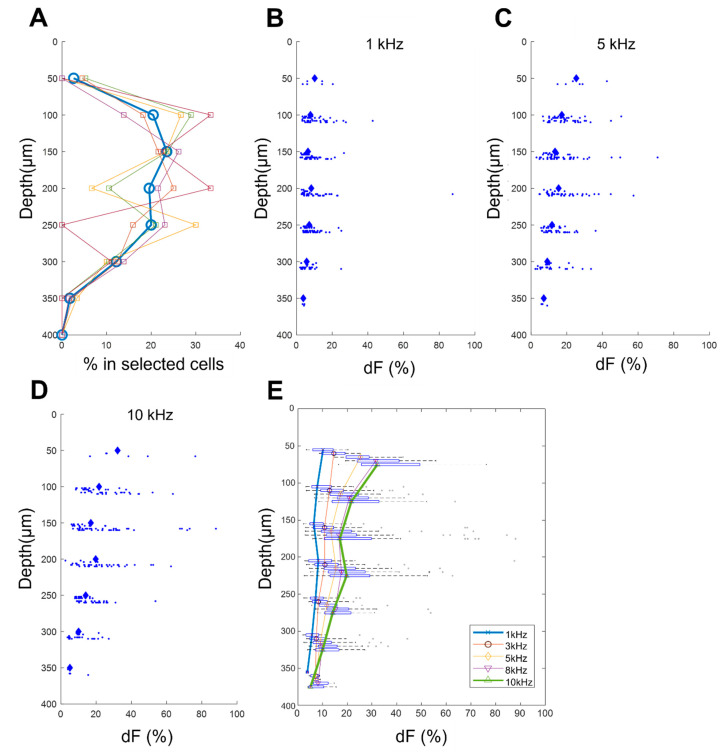
Effect of frequencies on neurons at different depths. (**A**) Distribution of the depth of selected neurons. Each line with a different color represents data from a different rat, and the thicker line represents the average of data from all five rats. The majority of neurons were situated at depths ranging from 100 to 300 µm. (**B**–**D**) Activation of neurons at varying depths during 1 kHz (**B**), 5 kHz (**C**), and 10 kHz (**D**) stimulations as examples. Median response for all rats is indicated by ◊, individual neurons by dots, and data from each rat are presented in separate horizontal columns. Neurons located closer to the surface displayed stronger activation compared to deeper neurons. (**E**) Neuronal activation during stimulation at frequencies ranging from 1 kHz to 10 kHz, depicted with median values and the 25th/75th percentile.

**Figure 5 biomedicines-12-01346-f005:**
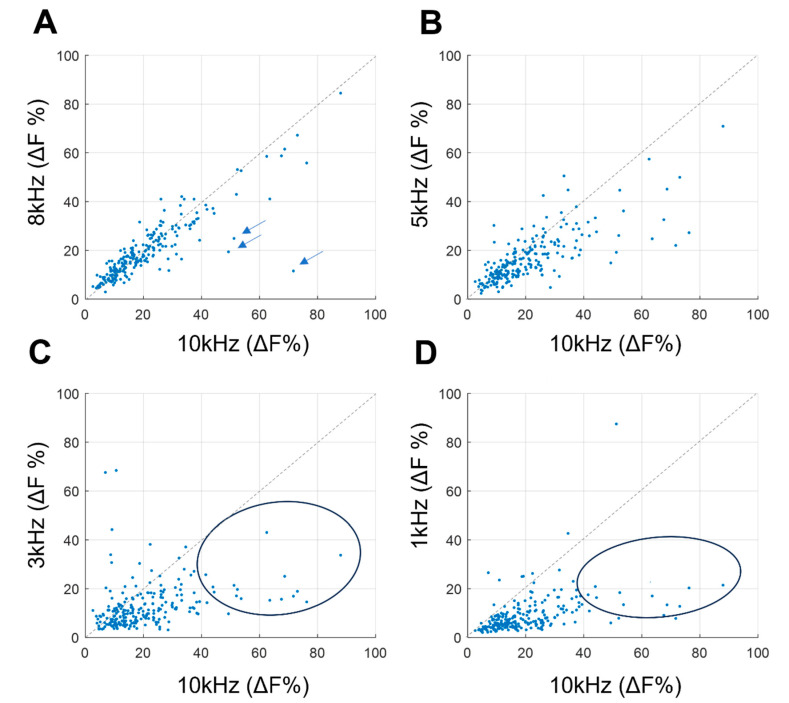
Comparison of dF in relation to 10 kHz stimulation. (**A**) Comparison between 10 kHz (*x*-axis) and 8 kHz (*y*-axis). A dashed diagonal line indicates equal response. The majority of neurons displayed a highly linear relationship and were activated by both 10 kHz and 8 kHz. A small group of neurons (indicated by the arrow) exhibited a stronger response to 10 kHz. (**B**) Comparison between 10 kHz (*x*-axis) and 5 kHz (*y*-axis). The linear relationship with 5 kHz was weaker compared to the response seen in (**A**). (**C**) Comparison between 10 kHz (*x*-axis) and 3 kHz (*y*-axis). (**D**) Comparison between 10 kHz (*x*-axis) and 1 kHz (*y*-axis). As the frequency decreased (as observed in (**C**) and (**D**) at 3 kHz and 1 kHz), neurons that were weak responders to 1 and 3 kHz (<20% on the *y*-axis in (**C**,**D**)) exhibited a wider range of responses to 10 kHz, spanning from 0% to 40% on the *x*-axis.

**Figure 6 biomedicines-12-01346-f006:**
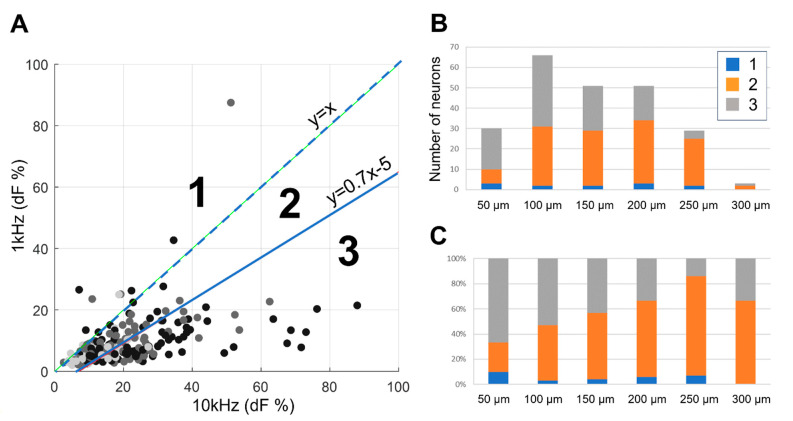
Frequency preference between 1 kHz vs. 10 kHz. (**A**) Response to 1 kHz vs. 10 kHz stimulation was categorized into three preference areas: (1) neurons activated more by 1 kHz, (2) neurons mildly preferring 10 kHz, and (3) neurons highly preferring 10 kHz, distinguished by two lines (line between area 2 and 3 are somewhat arbitrary, but qualitative results did not change). (**B**) Distribution of the number of neurons at different depths for each frequency preference area. The major population for analysis, spanning depths from 100 to 200 µm, exhibited mild (2) or strong (3) preference for 10 kHz. (**C**) Percentage of neurons in each preferred area. Superficial neurons (at a depth of 50 µm) demonstrated a strong (or mild) preference for 10 kHz, and this preference decreased as neurons were located deeper. The percentage of mild preference increased for deeper neurons (e.g., at 250 µm). A small population of neurons exhibited a preference for 1 kHz in superficial layers, although this preference remained under 10% for neurons at any depth.

**Figure 7 biomedicines-12-01346-f007:**
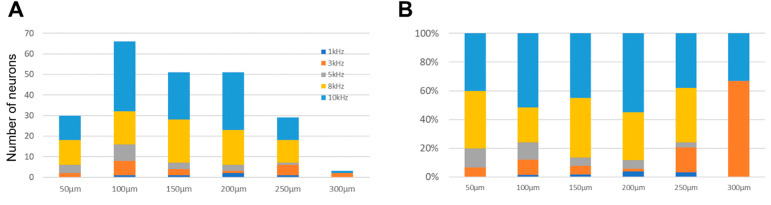
Optimal frequency for neuronal activation at various depths. (**A**) Number of neurons activated by the best frequency at depth. (**B**) percentage of neurons categorized by the best frequencies for activation.

## Data Availability

The data is not accessible to the public because of intellectual property rights.

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
