# Peer review of "Frequency-Dependent Neural Modulation of Dorsal Horn Neurons by Kilohertz Spinal Cord Stimulation in Rats"

_biomedicines, 2024, doi:10.3390/biomedicines12061346_

Round 1

Reviewer 1 Report

Comments and Suggestions for Authors

The present article shows the implications of frequency-dependent neuronal modulation of dorsal horn Neurons by Kilohertz Spinal Cord Stimulation in Rats . The topic is relevant, but the major deficiencies identified in both content and form need to be addressed based on the specific recommendations below:

1. I recommend that the instructions for authors provided by the journal be rechecked and adapted in terms of authors' information on the first page.

2. The conclusion part of the abstract should be improved in terms of results and future research directions to which this research can be referred.

3. The introduction section should immediately follow the keywords as in the template provided by the journal for authors.

4. The mechanism of action underlying the kHF-SCS should be detailed and updated with what is currently known even if it is not fully elucidated because the introduction and poorly presented in relation to the complexity of the topic.

5. The scope of the paper should be improved in terms of describing the contribution to the field under review and the elements of scientific novelty presented because the authors only presented in the last paragraph of the introduction what they did in the study.

6. Sections and subsections are numbered according to the instructions for authors.

7. It is important to explain why only male rat pups were chosen.

8. The title of Figure 6 should be placed below the figure.

9. How was the study method validated?

10. The discussion section should be further detailed by comparing the results obtained with those of other studies.

11. A sub-section on future research directions should be created or included in the summary of conclusions.

12. A number and date of a document attesting to the ethical validation of the present study should be provided.

Author Response

  1. I recommend that the instructions for authors provided by the journal be rechecked and adapted in terms of authors' information on the first page.

Thank you for your recommendation. We reviewed the guidelines and made any necessary adjustments to ensure clarity and accuracy regarding authors' information on the first page.

  1. The conclusion part of the abstract should be improved in terms of results and future research directions to which this research can be referred.

Thank you for your feedback. We have revised the conclusion portion accordingly to better reflect the outcomes and potential avenues for further investigation.

[L411] Building upon the findings of this study, further investigation is warranted to elucidate the specific mechanisms underlying the observed correlation between higher stimulation frequencies (1 to 10 kHz) and increased activation of dorsal horn neurons. Additionally, exploring how different frequencies influence the processing of sensory and pain-related information within the spinal network could provide valuable insights into optimizing kHZ-SCS parameters for enhanced pain relief. Moreover, investigating the long-term effects of frequency-dependent modulation on dorsal horn neuron activity and the potential for neuroplastic changes in response to chronic kHZ-SCS therapy would be instrumental in refining treatment strategies for chronic pain management. Overall, future research directions should aim to translate these findings into clinically relevant interventions that improve the efficacy and durability of kHZ-SCS-based therapies for individuals suffering from chronic pain conditions.

  1. The introduction section should immediately follow the keywords as in the template provided by the journal for authors.

We have adjusted the manuscript so that the introduction section promptly follows the keywords, as per the provided template for authors. [L22]

  1. The mechanism of action underlying the kHF-SCS should be detailed and updated with what is currently known even if it is not fully elucidated because the introduction and poorly presented in relation to the complexity of the topic.

We have revised the introduction to ensure that it adequately reflects the current knowledge on this complex topic and provides a clearer overview of the underlying mechanisms involved in kHF-SCS. However, there were not many publications proposed paresthesia-free pain relief mechanism of 10kHz SCS.

[L37] The mechanism of action underlying kHF-SCS is complex and not yet fully elucidated but is believed to involve the recruitment of inhibitory pathways within the spinal cord [5,12,13]. Recent studies using a rat model of neuropathic pain suggested that kHF-SCS may promote the release of endogenous opioids as part of its pain-relieving mechanism [14] or potentially reduces hyperalgesia by enhancing autophagic activity within the spinal cord[15]. Moreover, kHF-SCS may modulate spinal cord neuronal excitability and synaptic transmission, ultimately leading to a reduction in pain signals reaching the brain [16].

  1. The scope of the paper should be improved in terms of describing the contribution to the field under review and the elements of scientific novelty presented because the authors only presented in the last paragraph of the introduction what they did in the study.

 Thank you for your feedback. Enhancing the scope of the paper to clearly articulate its contribution to the field and highlight the elements of scientific novelty is crucial for contextualizing the significance of the study. We have updated the introduction to provide a more comprehensive overview of the research's contribution and the novel aspects introduced, ensuring that these aspects are appropriately emphasized throughout the manuscript.

[L43] Stimulation of spinal cord dorsal horn neurons through kHF-SCS is a key mechanism in pain management. The dorsal horn of the spinal cord is integral to processing pain signals, where sensory information from peripheral nerves is transmitted to higher brain centers[17]. By targeting specific regions within the dorsal horn, kHF-SCS can inhibit the transmission of nociceptive signals while promoting the activation of inhibitory pathways, ultimately leading to pain relief. Understanding the precise effects of kHF-SCS on dorsal horn neurons activation is crucial for optimizing treatment outcomes and developing more effective pain management strategies for individuals with chronic pain conditions.

  1. Sections and subsections are numbered according to the instructions for authors.

Thank you for your observation. We have changed the format and the sections and subsections are correctly numbered according to the provided instructions for authors.

  1. It is important to explain why only male rat pups were chosen.

Thank you for raising this point. The selection of male rat pups was based on several factors, including the need to minimize variability in hormonal fluctuations associated with the estrous cycle in female rats. Additionally, previous research has shown that male rats often exhibit more consistent behavioral responses in certain experimental paradigms, which can enhance the reliability and reproducibility of the study findings.

  1. The title of Figure 6 should be placed below the figure.

We have corrected this issue in the revised manuscript.

  1. How was the study method validated?

The calcium imaging method employed in the study was validated through several steps to ensure its reliability and accuracy in capturing neuronal activity [Reference:19]. Firstly, prior to conducting the experiments, the calcium indicator dye used for imaging was tested to confirm its specificity in detecting changes in intracellular calcium levels, which serve as a proxy for neuronal activation.

During the experiments, control trials were conducted to verify the baseline activity of dorsal horn neurons before any stimulation was applied. This baseline activity served as a reference point for comparison with the activity observed during and after kilohertz high-frequency spinal cord stimulation (kHF-SCS).

Furthermore, the calcium imaging setup and procedures were validated through pilot experiments and control conditions to confirm the consistency and reproducibility of the results. This involved assessing factors such as signal-to-noise ratio, imaging resolution, and the stability of the calcium indicator dye over time.

Additionally, the data obtained from calcium imaging were cross-validated electrophysiological recordings to corroborate the findings and ensure their reliability.

Overall, rigorous validation of the calcium imaging method was integral to the study, and appropriate controls and validation steps were implemented to ensure the accuracy and validity of the results.

  1. The discussion section should be further detailed by comparing the results obtained with those of other studies.

 Thank you for your feedback. We have enhanced the discussion by incorporating comparisons with relevant research, highlighting similarities, differences, and potential implications for the field. This will provide a more comprehensive understanding of how our results contribute to the existing body of knowledge on the topic.

[L371-379] We briefly discussed the comparison and difference from Lee at al and Sagalajev et al in the limitations.

  1. A sub-section on future research directions should be created or included in the summary of conclusions.

Thank you for your suggestion. We have incorporated potential avenues for further investigation, highlighting areas where additional research could contribute to advancing our understanding of the topic and its implications in the conclusion part.

[L414] Moreover, investigating the long-term effects of frequency-dependent modulation on dorsal horn neuron activity and the potential for neuroplastic changes in response to chronic kHZ-SCS therapy would be instrumental in refining treatment strategies for chronic pain management. Overall, future research directions should aim to translate these findings into clinically relevant interventions that improve the efficacy and durability of kHZ-SCS-based therapies for individuals suffering from chronic pain conditions.

  1. A number and date of a document attesting to the ethical validation of the present study should be provided.

We acknowledge the importance of providing documentation attesting to the ethical approval of our research. We have included the specific number and date of the document validating the ethical considerations of our study in the revised manuscript.

[L55] All experimental procedures were conducted in accordance with animal care guidelines approved by the Institutional Animal Care and Use Committee (IACUC) at Explora BioLabs (San Diego, CA) (#EB20-016-001, Jan/11/2021).

Reviewer 2 Report

Comments and Suggestions for Authors

The study presented is well designed, written and with clear objectives. The hypothesis about the possible specificity of parameters when performing spinal electrical stimulation constitutes an objective that has been of interest to the scientific community in recent years.

The fact of associating electrical stimulation in the dorsal horn of rats with the treatment of clinical symptoms associated with chronic pain is quite ambitious. In my opinion, we suggest authors to be more cautious in their statements. If they want to compare possible applications of this intervention in humans, it is necessary for the authors to justify the current knowledge gap in the clinical application of electrical stimulation of the dorsal horn in people with chronic pain.

It would be interesting if the authors showed in a figure the electrical stimulation device used and its placement in the sample.

Finally, the bibliography section must be reviewed in its formal aspects.

Congratulations to the authors for the nice study presented!

Author Response

  1. The study presented is well designed, written and with clear objectives. The hypothesis about the possible specificity of parameters when performing spinal electrical stimulation constitutes an objective that has been of interest to the scientific community in recent years.

We appreciate the reviewer’s recognition of the clarity of our objectives and the relevance of our hypothesis regarding the specificity of parameters in spinal electrical stimulation. Indeed, this aspect has garnered significant interest within the scientific community in recent years, and we aimed to contribute valuable insights through our research. We are pleased to hear that our study design and writing were well-received, and we remain committed to advancing knowledge in this important area of research.

  1. The fact of associating electrical stimulation in the dorsal horn of rats with the treatment of clinical symptoms associated with chronic pain is quite ambitious. In my opinion, we suggest authors to be more cautious in their statements. If they want to compare possible applications of this intervention in humans, it is necessary for the authors to justify the current knowledge gap in the clinical application of electrical stimulation of the dorsal horn in people with chronic pain.

Thank you for your insightful comment and consideration regarding the ambitious nature of associating electrical stimulation in the dorsal horn of rats with the treatment of clinical symptoms associated with chronic pain. We acknowledge the need for caution in our statements and the importance of justifying any comparisons to potential applications in humans.  In the revised version, we provided a more nuanced and evidence-based discussion of the translational implications of our findings, while also highlighting areas for further research and potential challenges in extrapolating results from animal models to human clinical settings.

[L410] Building upon the findings of this study, further investigation is warranted to elucidate the specific mechanisms underlying the observed correlation between higher stimulation frequencies (1 to 10 kHz) and increased activation of dorsal horn neurons. Additionally, exploring how different frequencies influence the processing of sensory and pain-related information within the spinal network could provide valuable insights into optimizing kHZ-SCS parameters for enhanced pain relief. Moreover, investigating the long-term effects of frequency-dependent modulation on dorsal horn neuron activity and the potential for neuroplastic changes in response to chronic kHZ-SCS therapy would be instrumental in refining treatment strategies for chronic pain management. Overall, future research directions should aim to translate these findings into clinically relevant interventions that improve the efficacy and durability of kHZ-SCS-based therapies for individuals suffering from chronic pain conditions.

  1. It would be interesting if the authors showed in a figure the electrical stimulation device used and its placement in the sample.

Thank you for your suggestion. We utilized a Nevro trial stimulator (TSM) in conjunction with a microelectrode to administer kilohertz spinal cord stimulation (kHz-SCS). The positioning of the stimulation electrode is illustrated in Figure 1A. For further details regarding the TSM, please refer to the Nevro website at https://nevro.com/English/us/providers/product-trial-stimulator/default.aspx

  1. Finally, the bibliography section must be reviewed in its formal aspects.

Thank you for bringing this to our attention. We reviewed the bibliography section to address any formal aspects that may require attention.

Reviewer 3 Report

Comments and Suggestions for Authors

The researchers concluded that higher kilohertz frequencies correlated with increased activation of dorsal horn neurons, in contrast to blocking. The findings of this study may have clinical relevance in the context of using kilohertz SCS for the management of chronic pain and treatment of various neurological disorders. Further research is required to elucidate the precise mechanisms and therapeutic applications of this deeper activation.

Discuss previous studies that have used kilohertz spinal cord stimulation for pain management and how their findings contribute to this body of knowledge.

The objectives are clearly stated and cover the main aspects of the research. They provide a comprehensive overview of what the study aims to achieve. The objectives are specific, measurable, achievable, relevant, and time-bound, which are the key characteristics of well-defined objectives. Therefore, I have no improvements to suggest in this area. However, clearly discuss how each objective was addressed by the results.

The methods section could provide more detail to ensure the replicability and reproducibility of the study.  provide more information about the selection criteria for the rat pups, such as their health status, weight, and any specific strain or genetic background.  provide more details about the dissection solution and recovery solution, including their composition and pH.  provide more details about the settings of the microscope and camera, such as the magnification used, exposure time, and image resolution. provide more details about the custom MATLAB functions used for image data processing and analysis.  provide the specific criteria used for determining significance among groups and between groups.

The discussion could provide even more detail and clarity about the strengths and limitations of the study. further emphasize the novelty and significance of your findings, particularly in the context of existing literature, and discuss how your study addresses gaps in the current understanding of kilohertz SCS. While you have acknowledged several limitations, provide more detail about how these limitations might have affected their results and conclusions. For example, discuss the potential impact of using rat pups on the generalizability of your findings to adult rats or other species. provide more specific suggestions for future research, such as potential experimental designs or methodologies that could address the limitations of their study.

explain how you determined the preference of neurons for different frequencies and how you quantified the activation of neurons.

Author Response

  1. Discuss previous studies that have used kilohertz spinal cord stimulation for pain management and how their findings contribute to this body of knowledge.

We appreciate the suggestion to discuss previous studies that have utilized kilohertz spinal cord stimulation (SCS) for pain management and their contributions to the existing body of knowledge. We have incorporated a detailed discussion of relevant literature in our manuscript, highlighting key findings from previous studies and elucidating how they inform and complement our own research. This will provide valuable context and enhance the comprehensiveness of our discussion on the topic.

[L371] Compared to our previous work [5] showing almost no activation at 1-5 kHz, there was noticeable response in current study. The main difference might be the previous paper used in vivo preparation where all rostro-caudal connections were intact, while the in vitro slice preparation from this work sacrifices those connections from crucial inhibitory neurons (ex. Islet cells with long rostro-caudal innervation) could modulate neighbor neurons [22].

This study could not confirm if 10kHz activates dorsal column or not on transverse slice, while Sagalajev et al [31] have suggested that kilohertz frequency can activate axons at 50% of motor threshold, which is considered above perception threshold in rat [32,33]. Data from our clinical studies have suggested that therapeutic amplitudes for kHz frequency SCS are no greater than 25% of the kHz paresthesia threshold. Thus, the conclusions of Sagalajev et al would relate to paresthesia-based kHz SCS, which is not used clinically.

  1. The objectives are clearly stated and cover the main aspects of the research. They provide a comprehensive overview of what the study aims to achieve. The objectives are specific, measurable, achievable, relevant, and time-bound, which are the key characteristics of well-defined objectives. Therefore, I have no improvements to suggest in this area. However, clearly discuss how each objective was addressed by the results.

Thank you for your thorough assessment of our objectives. We appreciate your acknowledgment of their clarity and alignment with the main aspects of the research. We have clearly discussed how each objective was addressed by the results [L126, L157, L196-199, L231, L245, L270] in our revised manuscript.

  1. The methods section could provide more detail to ensure the replicability and reproducibility of the study.  provide more information about the selection criteria for the rat pups, such as their health status, weight, and any specific strain or genetic background.  provide more details about the dissection solution and recovery solution, including their composition and pH.  provide more details about the settings of the microscope and camera, such as the magnification used, exposure time, and image resolution. provide more details about the custom MATLAB functions used for image data processing and analysis.  provide the specific criteria used for determining significance among groups and between groups.

Thank you for your detailed feedback on the methods section. We recognized the importance of providing sufficient detail to ensure the replicability and reproducibility of our study. Here are the additional details we have incorporated in the revised version.

  1. Selection criteria for rat pups: We included information on the health status, weight, strain, and genetic background of the rat pups used in the study to provide transparency and enable accurate replication.

[L56] Healthy Sprague Dawley rat pups, aged between 7 to 12 days (body weight 12-18g), were used in our experiments. These pups were housed with their mother and maintained on a 12-hour light/dark cycle in a temperature-controlled environment with unrestricted access to food and water. Typically, male rat pups were selected.

  1. Dissection and recovery solutions: We provided detailed information on the composition, pH, and preparation of the dissection and recovery solutions used in the experiments to ensure consistency and reproducibility.

[L63] and placed in oxygenated ice-cold dissection solution (in mM: 95 NaCl, 2.5 KCl, 1.25 NaH2PO4, 26 NaHCO3, 50 sucrose, 25 glucose, 6 MgCl2, 1.5 CaCl2, and 1 kynurenic acid, pH 7.4, 320 mOsm).

[L66] oxygenated recovery solution (in mM: 125 NaCl, 2.5 KCl, 1.25 NaH2PO4, 26 NaHCO3, 25 glucose, 6 MgCl2, and 1.5 CaCl2, pH 7.4, 320 mOsm) containing a fluorogenic calcium-sensitive dye (5 µM, Aat Bioquest Calbryte™ 520 AM, Cat# 20653) for 30 minutes at 35°C [19]

  1. Microscope and camera settings: We included specific details on the magnification, exposure time, and image resolution settings of the microscope and camera used for imaging to enable replication of the experimental setup.

[L84] Images were acquired by the Olympus CellSen software with a sampling rate of 2 Hz (Figure 1B) and a resolution of f 2048x2048 under a 40X magnification

  1. MATLAB functions: We provided a description of the custom MATLAB functions used for image data processing and analysis, including their purpose, implementation, and any relevant parameters.

[L87] Image data was processed and analyzed using custom MATLAB functions combined with Image processing toolbox. Individual cells were marked manually and pixels (5µmx5µm) around the marked point were used for Fluorescence data.  

  1. Criteria for determining significance: We outlined the specific criteria used for determining significance both within groups and between groups, including statistical methods and thresholds employed.

[L94] The initial step involved employing the Nonparametric Kruskal-Wallis one-way analysis of variance (ANOVA) to assess significance (p<0.01) among groups. Subsequently, a post-hoc two-sided Wilcoxon rank sum test, adjusted for Bonferroni correction (p<0.01), was utilized to determine the significance of differences between groups.

  1. The discussion could provide even more detail and clarity about the strengths and limitations of the study. further emphasize the novelty and significance of your findings, particularly in the context of existing literature, and discuss how your study addresses gaps in the current understanding of kilohertz SCS. While you have acknowledged several limitations, provide more detail about how these limitations might have affected their results and conclusions. For example, discuss the potential impact of using rat pups on the generalizability of your findings to adult rats or other species. provide more specific suggestions for future research, such as potential experimental designs or methodologies that could address the limitations of their study.

Thank you for your insightful feedback on the discussion section of our manuscript. We appreciate your suggestions for further enhancing the clarity and depth of our discussion. Here’s the solutions to address your points:

  1. Strengths and limitations: We provided a more comprehensive discussion of both the strengths and limitations of our study, including a detailed examination of how each limitation may have influenced our results and conclusions. Specifically, we discussed the potential impact of using rat pups on the generalizability of our findings to adult rats or other species, as well as any other factors that may have affected the robustness of our results. [Discussion: L344, L359, L366]
  2. Novelty and significance: We further emphasized the novelty and significance of our findings, particularly in the context of existing literature on kilohertz spinal cord stimulation. We highlighted how our study addresses gaps in the current understanding of kilohertz SCS and contributes to advancing knowledge in the field. [L387-391]
  3. Future research suggestions: We provided more specific suggestions for future research, including potential experimental designs or methodologies that could address the limitations of our study. [L350-352]
  4. explain how you determined the preference of neurons for different frequencies and how you quantified the activation of neurons.

We employed fluorescent calcium indicators that report changes in intracellular calcium levels, serving as a proxy for neuronal activity. Through calcium imaging, we observed frequency-dependent changes in neuronal activation patterns, allowing us to determine the preference of neurons for different frequencies which made the highest response in Fluorescence change [L272]. Additionally, calcium imaging facilitated the mapping of activated neurons across different depths within the dorsal horn, providing insights into the spatial distribution of neuronal activation [L273-277]. Overall, the application of calcium imaging techniques enhanced the precision and sensitivity of our experimental approach, enabling a comprehensive investigation into the frequency-dependent modulation of dorsal horn neuron activity in the context of pain processing.

Round 2

Reviewer 1 Report

Comments and Suggestions for Authors

The authors have significantly improved the manuscript based on the suggestions received.